# Decent Work, ILO’s Response to the Globalization of Working Life: Basic Concepts and Global Implementation with Special Reference to Occupational Health

**DOI:** 10.3390/ijerph17103351

**Published:** 2020-05-12

**Authors:** Jorma Rantanen, Franklin Muchiri, Suvi Lehtinen

**Affiliations:** 1Department of Public Health/Occupational Health, University of Helsinki, 00290 Helsinki, Finland; 2International Labour Office, 1211 Geneva, Switzerland; Muchiri@ilo.org; 3International Affairs, Finnish Institute of Occupational Health, FIOH, 00250 Helsinki, Finland; suvi.a.lehtinen@outlook.com

**Keywords:** Decent Work, globalization, ILO, occupational health, SDGs

## Abstract

Twenty years ago, the International Labour Organization (ILO) launched a new strategy, the Decent Work Agenda, to ensure human-oriented development in the globalization of working life and to provide an effective response to the challenges of globalization. We searched for and analysed the origin of the Decent Work concept and identified the key principles in ILO policy documents, survey reports, and relevant United Nations’ (UN) documents. We also analysed the implementation of the Decent Work Country Programmes (DWCPs) and examined the available external evaluation reports. Finally, we examined the objectives of the ILO Decent Work Agenda and the Decent Work targets in the UN 2030 Agenda for Sustainable Development in view of occupational health. In two thirds of the ILO’s Member States, the Decent Work Agenda has been successfully introduced and so far fully or partly implemented in their DWCPs. The sustainability of the Decent Work approach was ensured through the UN 2030 Agenda, the ILO Global Commission Report on the Future of Work, and the ILO Centenary Declaration. However, objectives in line with the ILO Convention No. 161 on Occupational Health Services were not found in the DWCPs. Although successful in numerous aspects in terms of the achievement of the Decent Work objectives and the UN Sustainable Development Goals (SDGs), the Decent Work Agenda and the Decent Work Country Programmes need further development and inclusion of the necessary strategies, objectives, and actions for occupational health services, particularly in view of the high burden of work-related diseases and, for example, the present global pandemic. In many countries, national capabilities for participation and implementation of Decent Work Country Programmes need strengthening.

## 1. Introduction

A total of 3.5 billion people—63% of the world’s population—belong to the global workforce. They spend more than one third of their adult life at work and produce a total world gross domestic product (GDP) of US$ 88 trillion [1], maintaining the social fabric and providing the material basis for living for nations, communities, families, and individuals. In spite of revolutionary technological development, the globalization of economies, digitalization, artificial intelligence, and robotization, human work continues to be the key factor behind the material and social well-being of all countries. Thus, the health and safety of workers and their work ability, competence, skill, and labour participation are the key factors of the socioeconomic development and sustainability of every country.

There is evidence of the positive impact of globalization on the global economic growth of the world as a whole, particularly on the growth of world trade and the improved distribution of wealth also to the developing world, for example by halving the global poverty rates in the past 20 years. A similar positive impact is seen in the distribution of new technologies and the Internet throughout the continents and enhancement of global connectivity.

The social impact of globalization, and on occupational health in particular, is, however, not universally positive. The transfer of basic and manufacturing industries to low-wage countries has generated many positive impacts there, but left behind major employment and social problems in the former host countries and communities. Still, most of the two billion workers of the developing world, of whom over 50% are informal, have not benefited from globalization; sometimes quite the opposite. In particular, agricultural and rural populations, family farmers, fishers, and forestry workers, domestic workers, and small and microenterprises in developing countries have been losers in globalization. Several vulnerable groups in both the industrialized and developing countries are also recognized as potential victims. Mr. Juan Somavia, Director General of the ILO, stated before the 87^th^ International Labour Conference: “Globalization has brought prosperity and inequalities, which are testing the limits of collective social responsibility” [2,3,4,5].

The ILO Decent Work Agenda is a unique global-scale response to the challenges of globalization, employment, economic, social protection policies, and social dialogue. This article describes the origin of the Decent Work concept and analyses its implementation at the national level, and particularly the role of occupational health as a part of the social protection pillar of the Decent Work Agenda [5].

## 2. Methods

This explorative literature review was drawn up by studying the origin of the Decent Work concept and its key principles in selected and relevant ILO policy documents, survey reports, and relevant UN documents (Appendix A). The selection of the documents was done by the authors with long-time follow-up of the ILO Decent Work policies. The implementation of the Decent Work Country Programmes (DWCPs) was examined from the available ILO documents. Finally, we examined the Decent Work targets in the ILO Decent Work Agenda and in the UN 2030 Agenda for Sustainable Development. A special focus was set on occupational health elements in the DWCPs. In addition to policy documents, important sources of information for the analysis were the ILO external high-level evaluation reports of subregional Decent Work Programmes and their imbedded national programme evaluations. The ILO evaluation criteria, relevance, coherence, effectiveness, efficiency, impact, sustainability, and overall performance were examined and used for the overall assessment and drawing our general conclusions on the realization through DWCPs of the Decent Work Agenda in selected subregions and their countries.

## 3. Decent Work–Basic Concepts

### Origin of the Decent Work Concept

In the 1980s, the pace of globalization was accelerated by several geopolitical changes (opening of global markets, particularly China) and the increasing wide-scale implementation of new information and communication technologies. Working conditions changed radically; enterprise structures were reorganized (downsizing, outsourcing, offshoring); and working contracts became fragmented (precarious work, zero-hour contracts, long and deviating working hours, agency work).

In 1998, a year before launching the Decent Work Country Programme (DWCP) approach, the 86th International Labour Conference (ILC) launched the Declaration on Fundamental Principles and Rights at Work [2], which was to apply to all working people in all 187 Member States of the ILO, regardless of the level of economic development and whether the Member States had ratified the fundamental conventions. The Declaration also covered groups with special needs, including the unemployed and migrant workers. The declared universal fundamental rights and principles were:Freedom of association and right to collective bargaining;Elimination of forced or compulsory labour;Abolition of child labour;Elimination of discrimination in employment and occupation.

The eight ILO Fundamental Conventions concern the freedom of association, the effective recognition of the right to collective bargaining, forced labour, the abolition of forced labour, the effective abolition of child labour, the minimum age convention, equal remuneration, and the elimination of discrimination with respect to employment.

By ratifying, the countries commit themselves to regulation and implementation in line with the Conventions. This commitment is supported by the ILO Follow-up procedure. The Member States that have not ratified one or more of the ILO Fundamental Conventions are asked each year to report on the status of the relevant rights and principles within their borders and to note any impediments to ratification and areas in which they may require assistance. These reports are reviewed by the Committee of Experts on the Application of Conventions and Recommendations (CEACR). Their observations in turn are considered by the ILO’s Governing Body.

As a response to globalization, the ILO developed the Decent Work approach to provide a social dimension to globalization and to initiate an intensive human-oriented approach to the challenges posed by the globalization of working life. In February 1999, the Secretary General of the UN, Mr. Kofi Annan, in introducing the new UN concept of the Global Compact before the World Economic Forum [3], emphasized the importance of shared values and principles in managing the challenges of globalization. “Unless globalization works for all, it will work for nobody,” he stated. He called for policies that give a human face to the global market and invited business leaders to “embrace, support and enact a set of core values in the areas of human rights, labour standards, and environmental practices”. In June 1999, the Director General of the ILO, Mr. Juan Somavia, in his Report to the 87th ILC, responded to this appeal by presenting an extensive document on Decent Work (Figure 1) [4,5]. According to the Director General, Decent Work is defined by the ILO as “productive work for women and men in conditions of freedom, equity, security and human dignity” 

The ILO’s Decent Work Agenda promotes a development strategy that recognizes the central role of work in everyone′s life. The Organization provides support in the form of integrated Decent Work programmes developed at the country level with ILO′s constituents. These programmes set priorities and targets within national development frameworks and aim to tackle major Decent Work shortcomings through effective programmes that also meet each of ILO′s four strategic objectives:to promote and implement the standards and fundamental principles and rights at work;to enhance opportunities for men and women to obtain decent employment and wages;to expand the scope and heighten the effectiveness of social protection for all;to strengthen tripartism and social dialogue.

In addition, Mr. Somavia in his address to the 87th ILC recalled the issues inherent in the concept of Decent Work and outlined the future role of ILO: “The ILO is concerned with Decent Work. The goal is not just the creation of jobs, but the creation of jobs of acceptable quality. The quantity of employment cannot be divorced from its quality. All societies have a notion of Decent Work, but the quality of employment can mean many things. It could relate to different forms of work, and also to different conditions of work, as well as feelings of value and satisfaction. The need today is to devise social and economic systems which ensure basic security and employment while remaining capable of adaptation to rapidly changing circumstances in a highly competitive global market” [4].

Later in 2008, the 97th Session of the ILC adopted the ILO Declaration on Social Justice for a Fair Globalization [6], which is implemented in the context of the Decent Work Agenda and its four strategic pillars: rights at work, employment creation and enterprise development, social protection, and social dialogue.

Since 1999, the ILO has initiated large numbers of DWCPs, which have also been evaluated. The ILO Secretariat regularly reports on the development and implementation of the Decent Work programmes to the ILO Governing Body and the ILC, at which Member States of the ILO meet in June every year in Geneva, Switzerland. This ensures regular follow-up by the highest governance levels of the Organization.

In the course of the globalization process, the relevance of and the need for the Decent Work Agenda have become even more important amid the widespread uncertainty in the world of work, from the financial turmoil and economic downturn of the last two decades to growing unemployment, informal employment, insufficient social protection, and labour migration. These have challenged governments and social partners—workers’ and employers’ organizations—to adopt the Declaration on Social Justice (for fair globalization) to strengthen the ILO’s capacity to promote its Decent Work Agenda and forge an effective response to the growing challenges of globalization.

Both the Decent Work Agenda and its implementation and, at the time, the UN Global Strategy with its Millennium Development Goals (MDGs) aim to eradicate extreme poverty and hunger through achieving full and productive employment and Decent Work for all, including women and young people (MDG Target 1b). This target highlights the fact that, despite progress, almost half of the world’s employed people are still working in vulnerable conditions, especially in Sub-Saharan Africa and Southern Asia. The MDGs succeeded in halving absolute poverty in the target period [7]. In spite of such significant success, according to ILO, more than 204 million people remained unemployed in 2015. This was over 34 million more than before the start of the 2008 economic crisis, and 53 million more than in 1991.

In 2015, the UN launched the new Global Strategy, Transforming our world: the 2030 Agenda for Sustainable Development, which contains 17 Sustainable Development Goals (SDGs) with hundreds of targets and actions addressing people, planet, prosperity, peace, and partnership [8].

The SDG strategy contains several goals, targets, and actions relevant to workers’ health and safety, particularly SDG No. 1 on the Elimination of poverty, SDG No. 3 on Health, and SDG No. 8 on the Promotion of inclusive and sustainable economic growth, full and productive employment and decent work for all. The ILO has been appointed the custodian of 13 different SDG indicators [9]. The UN, through the SDGs, recognizes the central role of Decent Work in enhancing the global sustainable development agenda. Thus, the UN Agenda will extend the life of the Decent Work Agenda until 2030, and likely beyond.

## 4. Decent Work Country Programmes

### 4.1. Background

The ILO launched the Decent Work concept in 1999, and the first DWCPs were begun in 2000 in the Philippines and the Ukraine. The DWCPs were intended to be well coordinated at the national level with other ILO and UN programmes on the development of working life, in order to make as effective use of scarce resources as possible. The DWCPs aimed at the distinct ILO contribution to UN country programmes and constituting one main instrument to better integrate regular budget and extra-budgetary technical cooperation [10]. The ILO biennial programme outcomes were designed to align well with the SDGs, enabling the ILO Global Technical Teams, field structures, flagship programmes, Centenary Initiatives, and DWCPs to work together and within the UN system to support Member States [11]. Decent Work called for quality jobs, dignity, equality, a fair income, and safe and healthy working conditions and environments; it strived to put people at the centre of development and create a future that is inclusive and sustainable [12].

### 4.2. Preparing the DWCPs

There has been long-term, logical continuity in the drafting and development of the DWCPs and the coordination, implementation, and independent evaluation of the accomplishments of the programmes. In order to facilitate the preparation of the DWCPs, in 2015, the ILO published a guideline for country-level actors’ support [10]. This guideline aims to provide a well-informed, comprehensive, but short diagnostic narrative of the growth, productive employment, and Decent Work situation and trends of each country. It provides the ILO constituents and other national stakeholders with coherent data on and analysis of the progress and situation related to Decent Work in each country. At the same time, it recognizes the key Decent Work challenges facing the country. The analytical report of the country offers data for the national development discourse. The analysis of the country situation can also be used as a basis for national training, capacity planning, and building for constituents and other key stakeholders [13,14] (Figure 2).

The country diagnostics provide an overall national development framework with demographics and data related to education, health and human development, the structure and performance of the economy, aspects related to inequality, vulnerability, and poverty in the country, employment and the labour market, the labour force, fundamental principles and rights at work, the implementation of international labour standards, decent working conditions and occupational safety and health (OSH), equal opportunities and treatment in employment, and questions related to social protection and social dialogue, to help identify and recognize the main Decent Work challenges ahead.

The logic of the DWCP process of the ILO is described in Figure 3.

On the basis of the diagnostics of the country situation, the national stakeholders, in collaboration with ILO experts, will draft a DWCP document to be discussed and agreed on in the tripartite process of the country. The process also serves as a learning cycle for the development of Decent Work. Examples of the main contents of the DWCPs are poverty reduction through the creation of Decent Work opportunities with a focus on young men and women, the reduction of child labour and elimination of its worst forms, or more and better employment for vulnerable groups, and responding to HIV and AIDS challenges in the world of work. The selected priorities depend on the deficits recognized in the Decent Work diagnosis of the country [15].

The DWCP process since the beginning of the DWCP implementation from 2000 to September 2019 is summarized by region in Table 1, showing a total of 121 countries’ participation.

The actual situation in September 2019 shows that Africa has the highest number of ongoing DWCPs, followed by Asia. Africa’s activities are even more prominent if the number of countries and the size of the workforce are considered (Table 2). A total of 41 programmes have been drafted, and 51 have already been approved by the Regional Director of the ILO. Regional Decent Work Programmes (DWPs) have also been adopted through regional meetings such as the DW Programme for Africa 2007–2015 and the Decent Work Programme for the Southern African Development Community (SADC).

The independent evaluations of the national DWCPs have been carried out at the sub-national, national, and regional levels, and the ILO has published the lessons learned to facilitate the further planning of new programmes. Evaluations have been carried out of both the individual country programmes and the regional programmes, such as those of the Arab Region (Jordan, Lebanon, the Occupied Palestinian Territory OPT), North Africa, the Caribbean, the Western Balkans, and Mekong Sub-region. These evaluations have identified success factors and needs for development and provided guidance for further actions [18].

## 5. Decent Work and Occupational Health

One of the fundamental principles of the ILO Constitution is the “protection of the worker against sickness, disease and injury arising out of his employment” [19]. In adopting the Declaration on Social Justice for a Fair Globalization [6], the ILO institutionalized the Decent Work concept, which had been earlier adopted at the 87th ILO Conference [4] and reaffirmed the continuation of OSH as the basic principle of Decent Work. It also established the InFocus Programme on SafeWork. This was further reinforced by the adoption of the ILO Centenary Declaration [20], confirming OSH as a fundamental element of Decent Work.

In 2003, the 91st ILC adopted the Conclusions of the Global Strategy on Occupational Safety and Health. These Conclusions confirmed OSH as key principles of Decent Work: “Decent Work must be safe work” [21]. The Conclusions pinpointed the need for tripartite national commitment and national action in fostering a preventive approach and a safety culture, which are key to achieving lasting improvements in safety and health at work.

In addition to the DW policy principles, the ILO has also produced legal instruments for the implementation of Decent Work in occupational health practice. Already before the 87th ILC (and Convention No. 161), Recommendation No. 112 (R112) on Occupational Health Services from 1959 guided the Member States in the organization of occupational health services (OHS) for their workers. In 1985, the ILC adopted the Occupational Health Services Convention No. 161 (C161) [22] and the related Recommendation No. 171 on Occupational Health Services [23], which superseded R112. These instruments guide the Member States in the development of national policies and corporate policies, laws and regulations, human resources and service infrastructures, and activities for the development of a national occupational health service, the OHS system. C161 has been ratified by 33 countries, and several countries have reported on its use as guidance for the development of OHS without ratification [24]. In 2017, the ILO Standards Review Mechanism-Technical Working Group established by the Governing Body reviewed C161 and R171 on Occupational Health Services, classified them as up-to-date standards, recommended specific promotion of C161 as a practical and time-bound follow-up action, and requested to be regularly updated in this respect [25].

ILO’s Convention No. 187 on the Promotional Framework for Occupational Safety and Health [26] emphasizes the importance of building national OSH systems. Item 3 of Article 4 of C187 lists OHS as an element of the national OSH system.

Recently, new evidence on the growing need for occupational health and OHS has accumulated. Globally, occupational diseases, occupational injuries, and work-related diseases (WRDs) are estimated to cause 2.8 million fatalities a year and over 300 million cases of non-fatal health outcomes that affect the health and work ability of workers, leading to a 4% loss of GDPs on average. Occupational injuries lead to 380,500 fatalities annually, whereas the mortality from WRDs is estimated at 2.41 million a year [27]. The cost estimates are based on the calculation of disability adjusted life years (DALY) from mortality and non-fatal health events. The majority of the estimated losses take place in the poorest countries and among the poorest and most vulnerable workers. In the process of globalization, particularly in rapidly growing and emerging economies, working people are increasingly vulnerable and have no access to OHS [24].

The trend of WRDs is expected to grow as working populations are ageing and numerous vulnerable groups such as migrant and refugee workers, several types of workers with vulnerabilities, and workers with partial work ability are recruited into the workforce and the number of informal workers grows. For example, half of the one billion World Health Organization (WHO) Western Pacific Region’s (WPRO) workers are informal. The informal status of workers is the key determinant of the absence of health, safety, and workers’ rights at work. The informal workers work in high-risk jobs, are exposed to high numbers of chemical, physical, biological, and social hazards, and are most vulnerable, for example, to infectious epidemics [27,28,29,30].

## 6. Challenges for Occupational Health and OHS in Decent Work

The SDGs aim among other things to eliminate poverty. In the long term, the only sustainable way out of poverty for countries, communities, families, and individuals is the gainful employment of working-age people. Successful employment needs job opportunities at the systems level and good employability at the individual level. A prerequisite for individual employability is good work ability (including health, functional capacity, competence, skill, and motivation), decent working conditions, and safety and health at work. The SDGs cannot be achieved without good health and work ability in working populations, as all the resources available for societies are derived from productive work. Some countries have included or are in the process of including the objectives of the development of occupational health, work ability, and OHS in their National Health 2030 strategies, thus harnessing occupational health for achieving the SDGs.

Some countries have achieved reasonably good working conditions and work environments, as well as occupational health, although the majority of countries still face great challenges in meeting the international standards set out in the Occupational Health Services Convention, C161. All countries need to exert a great deal of effort to achieve the UN’s SDGs in the field of health, safety, and work ability of workers. The greatest challenges lie in the improvement of the occupational health and work ability of workers in micro- and small-scale enterprises, the self-employed, and workers in the informal sector, who comprise the majority of the world’s workforce. Today, the majority of the workers of the world are still severely underserved and vulnerable. The gradual extension of the coverage and content of comprehensive OHS, including prevention, protection, promotion of health and work ability, care, rehabilitation, compensation in cases of injuries, and services for return to work is needed for all workers. Workers not yet covered, such as those in agriculture, the informal sector, and the self-employed, have the most urgent need for these services [24,27,30].

All occupational health and safety hazards can be prevented. Economics researchers have found the control of occupational accidents and diseases and the improvement of working conditions and occupational safety and health to be financially productive and profitable mainly through:(a)the reduction of economic loss by preventing the loss of work ability through accidents, diseases, presenteeism, disability, and sickness absenteeism and(b)increased productivity derived from better work ability, better motivation, and smoother flow of production (i.e., the ergonomic principle).

The high coverage, comprehensive content, and good functioning of OHS have been positively associated with the human development index (HDI) of the United Nations Development Programme (UNDP), the competitiveness index (World Economic Forum) and GDP per capita of countries, and the success and sustainability of enterprises [24].

Work does not produce only economic and material assets; participation in working life enables important social contacts, support from the work community, opportunities for vocational and professional development, and gives the individual a place and status in working organizations, institutions, living communities, and society at large.

Work itself is an important determinant of workers’ health, providing potential enhancement of health and work ability in good jobs and adverse effects in poor working conditions. Workers’ health and work ability also have a crucial impact on the health of families. In many societies, labour participation is the prerequisite for registration in social security schemes, which for their part reduce the risk of poverty. Still today, job opportunities and employment conditions vary greatly between and within countries and between different groups of workers and have great inequities among working people [31,32,33].

OSH has been a central issue for the ILO ever since its creation in 1919 and continues to be a fundamental requirement for achieving the objectives of the Decent Work Agenda. In reaffirming this, the ILO Centenary Declaration for the Future of Work, adopted by the ILO Conference at its 108^th^ session in Geneva in June 2019, stated “Safe and healthy working conditions are fundamental to decent work” [20]. With respect to OSH, the DWCPs have provided a key avenue for including OSH objectives as part of national development plans and for stimulating high-level decision-makers in governments to allocate resources for OSH [15].

Examination of the DWCPs have revealed that the contributions of occupational health and OHS have not been sufficiently utilized in national programmes. Even the total coverage of OHS in the world has remained low (18.5%). There may be several reasons for this, but the governance model of OHS is likely not the cause, as 73.5% of the respondent countries in International Commission on Occupational Health, ICOH’s OHS survey were governed by the ministry of labour alone or jointly by the ministry of labour and ministry of health [24].

Globally, the majority of work-related loss of DALYs is estimated on the basis of WRDs and only about 11% on the basis of occupational injuries [27]. The management and prevention of WRDs constitute a major challenge for the development of Decent Work. This challenge centres on a few WRDs: musculoskeletal disorders (MSDs) (15%), cardiovascular disorders (CVDs) (17.2%), and occupational cancer (12.4%). The rest, almost 50%, comes from “others”, mainly various types of WRDs. Diseases are responsible for almost 90% of all WRD-DALY loss. The ILO has earlier initiated important measures for identifying and recognizing occupational diseases by providing a list of them (a guidebook on diagnostic criteria is also under preparation), guiding workers’ health surveillance, and providing the ILO International Classification of Radiographs of Pneumoconiosis (now being finalized to include the digital standard). The Joint ILO-WHO Committee on Occupational Health has also initiated and implemented two WHO-ILO global programmes: the Global Programme for Elimination of Silicosis and the Global Programme for Elimination of Asbestos-Related Diseases (ARDs). The continuation of such policy trends fits well with the DWCPs. The practical prevention and management of the massive problem of WRDs, however, require the availability of comprehensive multi-professional OHS, i.e., universal occupational health coverage.

Including the development of OHS in the DWCPs would provide a strong substantive and practical contribution to the implementation of the DWCPs and make OHS available for underserved sectors as shown in the needs and provisions analysis in Table 3. The ILO is in the process of initiating a special campaign for the promotion of OSH conventions, including C161. This will also support the implementation of ILO’s C161 and also the achievement of the SDGs.

## 7. How to Develop Universal Occupational Health Coverage within the Framework of Decent Work

The WHO Global Strategy on Occupational Health for All (1995) called for universal OHS (“for all”) [31]. The United Nations General Assembly discussed universal health coverage (UHC) in its Sessions in 2012 and 2019 [34,35]. In May 2019, the World Health Assembly adopted three Resolutions, Nos. 72/2-4, on universal health coverage, aiming at the widest possible coverage of primary health care services for all people [36,37,38]. Universal health coverage is intended for the provision of basic general health for the total population, while universal occupational health coverage (UOHC) aims to cover working populations with the minimum of basic OHS. The occupational health of workers and the workplaces is not covered by general UHC, as occupational health requires paying attention to the specific needs of the workers’ work-specific health problems, as well as problems of work ability, the prevention of hazards and exposures at work and in the workplace, the management of working conditions and workloads, and the development of safe and healthy working conditions. The UN General Assembly in September 2019 requested among other things the provision of Decent Work and occupational health services for all, i.e., universality. Occupational health has three different groups of patients; workers, the workplace, and the working community at large. UOHC requires competence in occupational health, which is different from primary health care competencies of general practitioners (GPs) or family doctors. A unique feature of UOHC is that, like all other occupational health and safety issues, it is subject to tripartite collaboration between government authorities, employers, and workers.

The WHO Regional Office for the Western Pacific has supported the provision of UOHC in the Western Pacific Region [39,40]. The Strategy aims to fill the coverage gap in OHS in the WHO Region by focusing specifically on underserved and uncovered sectors, such as small and medium-sized enterprises (SMEs), the self-employed, and informal sector workers.

As stated by the ILO: “Decent Work is safe work” [5]. Workers’ health is a wider concept than traditional occupational safety only. It also contains health aspects, i.e., the prevention of health risks and hazards at work and in the workplace; the protection of workers against physical, chemical, biological, ergonomic, safety, psychological, and social factors hazardous to health; the promotion of workers’ health and work ability; the provision of health services to workers particularly in view of occupational and work-related diseases, acute health events, accidents, and emergencies; and guidance for workers in rehabilitation services and healthy working practices and healthy lifestyles. Pillar 4 of the Decent Work Agenda addresses occupational safety well, but occupational health less effectively. ILO’s C161 on OHS provides a valid international standard for the implementation of OHS within the Decent Work framework. The ILO-WHO-ICOH Basic Occupational Health Services approach provides a practical instrument for this implementation [41]. In particular, the new infectious epidemics and pandemics deserve more attention than before, which needs strengthening of occupational health services and in fact UOHC [37,38,39,40].

## 8. Globalization, Decent Work, and Future Challenges

The globalization process has continued over centuries, but during the last 30 years, it has developed and modified working life more intensively and faster than ever before. It is characterized by growing global trade and the movement of capital, materials, services, and people [42].

Globalization exposes all countries and enterprises to challenges and changes. Production structures will be radically modified; some branches of the economy will disappear; and new ones will appear. There will be changes in enterprise structures, their production methods, working practices, and work environments. Enterprises will need to adjust to several new trends; new work organizations, management systems, working contracts, working hours, OSH, and OHS. All of these mean that working life is expected to undergo major changes in terms of employment opportunities and requirements for competence, skills, and work ability [42,43,44,45,46].

In spite of progress in OSH in developed economies, the global situation is still unsatisfactory, with 2.8 million fatalities from occupational accidents and diseases a year and a manifold number of non-fatal injuries and diseases (see above). Many of the negative impacts of globalization hit the low-income and low and middle-income countries the worst, but they also affect advanced economies. Digital disruption affects enterprise and institutional structures. Some sectors of traditional economies are dying out, and their workers and professionals are at risk of exclusion from working life. Many traditional jobs are disappearing, and new challenging ones being generated; labour markets are being polarized; and in spite of improving averages, the inequities and gaps between the richest and the poorest are at risk of growing even further. New competence demands require new types of training, and numerous jobs are disappearing virtually overnight in large geographical areas, requiring movement of working populations to seek new job opportunities (particularly rural-urban migration). Downsizing, outsourcing, and offshoring in industries and services are leading to a new division of work between different countries and continents, which may affect the health and lives of working populations [47,48,49]. In particular, smaller countries, but also all national governments, need to adapt to global megatrends and develop new survival strategies. Policies, economies, production, and trade are being strongly dominated by giant multinationals.

With the help of new information and communication technologies, international interactions are being facilitated through ever-growing global connectivity and communication, including social media. Many other technological advancements, the introduction of new substances and materials including nanotechnologies and new e-technologies, digitalization, automation, and robotization in particular, and the use of artificial intelligence and big data are becoming effectively distributed throughout the global world of work. Automation and robotization are expected to eliminate 30–50% of current jobs, but new jobs are also being generated with different competence demands. Earlier, only physical and manual human power was replaced by automation, but the new trend is also partially replacing human intelligence by artificial intelligence: for example, speech recognition, problem-solving, learning, and planning.

The national and international division of work has changed due to technology transfer and the outsourcing, downsizing, and offshoring of industries and services. Global giant multinationals will dominate economic activities, exceeding the influence of national governments. Changes in production and in consumer markets will lead to a new division of work; some countries may become primarily producers of materials and producers of goods, others dealers of products only, and others only consumers with no large-scale production. This may result in occupational structures becoming one-sided and even more vulnerable to changes in global markets. National markets will mainly be transformed into global markets.

Due to robotization and artificial intelligence, the human role in production has become more one of planning, designing, quality control and services, transport and commercial, and marketing activities. The production of goods and materials will become assigned to automatic systems and robots. Some new branches of the economy will be generated, such as environmental and green economies, new energy economies, and knowledge economies. All these will have an impact on job contents, competences, and work ability profiles. [45,46,50,51,52].

Major changes have been observed in the demographic structures of the global workforce. Four of these macro trends are [53,54,55]:

The ageing of the workforce in most countries: In both the developing and industrialized countries, this is a dominant trend. Both developing and advanced economies show a comparatively rapid increase in the average age of working populations, known as the global ageing epidemic. This raises numerous challenges: the maintenance of decent employment for ageing individuals, the maintenance and promotion of their competence and work ability, the provision of employment opportunities, social protection, and the prevention of age discrimination. Better use of the competences and experience of ageing workers is warranted.

Youth: Some developing and emerging economies are facing challenges in the employment of younger age cohorts, which, particularly in Africa, has led to both internal (rural-urban migration) and international migration, particularly to Europe, but also elsewhere. The working conditions, OSH, training and education, fair remuneration, social protection, and overall life situation of young migrant workers is a huge international challenge.

The feminization of the workforce: In various parts of the world, this is progressing, providing many advantages for female workers, formal jobs, legal and social protection, better income, and more independence. Simultaneously, several jobs may have health hazards, long working hours, several psychosocial risks, reproductive hazards, balance of work and family life, and lack of gender equality and sexual harassment, calling for special protection of female workers at work.

Migrants, the handicapped, workers with partial work ability, and informal and domestic workers: These form a vulnerable group of workers and together constitute the majority of the global workforce. Their labour markets and social situation in global working life have not necessarily improved during the era of globalization, often in fact the opposite. International organizations, particularly the ILO, and non-governmental organizations (NGOs) have put great effort into the special protection of vulnerable groups of workers and made it a key element in Decent Work programmes.

Global economies are characterized by increased uncertainty of work contracts and employment; sudden changes in employment opportunities due to outsourcing and transfers of industries to new areas; quick, high peaks of employment, which may then dramatically drop; casual and temporary working contracts; and in general, higher risks of unemployment and insecure jobs. Such insecurity together with associated high job demands [48,56,57] and the high pace of global working life have caused psychological stress to become a global epidemic. General global challenges include growing global interdependence, climate change, and environmental degradation, which also modify the work environment, increase risks to workers’ health and safety, and affect families’ lives. Climate change challenges communities, enterprises, and private practices to respond through environmental protection, the greening of industries, and new consumer behaviour. These challenges also have an occupational health dimension in protecting workers against health hazards from climate and environmental hazards, such as excess solar radiation and air pollution, and in certain occupations, such as those of emergency response and rescue workers, against natural disasters. New green technologies also need to be carefully assessed for occupational health hazards [58,59,60].

Good work ability and psychological and physical functional capacity are key prerequisites for successful and productive employment. In the wider sense of the term, the concept of work ability is multidimensional and complex and includes workers’ personal health, psychological, cognitive, and physical capacity, knowledge, skills, and competence, attitudes and motivation to work, and personal social networks (particularly family). However, good work ability and functional capacity also require a healthy and safe work environment, a conducive working community, a secure labour contract, and social protection. Just and fair management supports work ability, as does the availability of OHS with service modules for work ability [46,61].

In the course of the development of working life, the challenges related to the occupational health and work ability of workers are gaining more importance. This is due to the growing psycho-social demands of work, the need to address the work ability problems of ageing workers, support and protection of the health of vulnerable groups, and consideration of the numerous health needs of migrant workers and other vulnerable groups. Maintaining and promoting the work ability of all working people requires decent working conditions, a safe and healthy work environment, physiologically and psychologically sound working time schedules, and limits of maximum working hours. This all requires well-functioning OHS for all working people in all types of jobs and enterprises and all modes of working contracts and employment models, i.e., UOHC.

The SDGs, particularly SDG 8, Decent Work for All, constitute a global response to the challenges of globalization by all sectors of societies and all regions of the world [8]. The Global Guarantee proposed by the ILO Global Commission in 2019 continues to humanize the further development of working life in the new era: “All workers, regardless of their contractual arrangement or employment status, should enjoy fundamental workers’ rights, an ‘adequate living wage’, maximum limits on working hours and protection of safety and health at work. Collective agreements or laws and regulations can raise this protection floor. This proposal also allows for safety and health at work to be recognized as a fundamental principle and right at work.” [19,20,46].

OSH and labour inspection (which are critically important for Decent Work) have been relatively well introduced as elements of Decent Work and are even further emphasized by Goal No. 8 of the UN Strategy on Sustainable Development, the ILO Global Commission, and the ILO Centenary Declaration [8,20]. In spite of the fact that about 89% of all occupational fatalities and a vast majority of non-fatal cases are attributed to WRDs and about 11% to occupational accidents, the occupational health dimension, however, has remained in the shadows in both the ILO Decent Work provisions and other national and international OSH policies. Recently, the WHO and UN provisions for UHC considered the scaling up of OHS [35,36,37,38]. All current and future trends, however, indicate a growing need for an occupational health approach in order to provide an effective response to the challenges of globalizing working life and to maintain the social fabric through the health and work ability of working people. In addition to all the regulatory, inspection, and control strategies, well-developed OHS are also needed to achieve the SDG for Decent Work. It is likely that this SDG’s objectives will not be achieved without the provision of UOHC for all working people.

## 9. External Evaluations of DWCPs

Each year, the ILO’s Evaluation Office holds consultations with senior management, the Evaluation Advisory Committee, and constituents to select topics for future high-level evaluations. The selected topics are then presented to the Governing Body (GB) for approval [62].

A total of 53 programmes of 121 DWCPs have so far been evaluated by external evaluators, as either regional or subregional or individual country evaluations using pre-set criteria [14,62,63,64,65]. The information needed for the DWCP evaluations was collected from country case studies and country missions and both group and individual interviews of national stakeholders. The collected data and information were subjected to systematic analysis using inductive, deductive (according to the ILO’s results-based management (RBM) approach), or abductive reasoning [65,66,67,68,69]. Triangulation between sources and methods was the primary method for ensuring the credibility of the findings and conclusions, and it was complemented by the stakeholders’ comments on the first draft of the evaluation report.

The following parameters were standardized for all DWCP evaluations:Relevance to the ILO DW agenda, DW/CPO, and country needsDesign, coherence, and validityEffectivenessEfficiencyImpactSustainability

Data from national DWCP reports were used in regional and sub-regional analyses. Evaluation projects during the period 2006–2018 are presented in Appendix A. The regional evaluation projects were implemented at different times, depending on the time schedules of the national programmes. Below are four concise descriptions of regional evaluations as examples of evaluation efforts.

### 9.1. Independent Evaluation of the ILO’s DWCP, Strategies and Actions in the Caribbean (2010–2015) 

The Caribbean region, with twenty countries of diverse sizes, varying from 5900 to three million citizens, is home for a total of approximately 60 million people. It has a young population structure (60% below 30) and is multi-ethnic and multi-lingual, mainly English-, French-, and Dutch-speaking, and its population is highly literate and well-educated.

The Caribbean economies are mainly based on agriculture, sugar, and spices and related rum distillation, but are complemented to a growing extent by light industries, electronics, textiles, handicrafts, and tourism. Some countries also have a notable mining industry. Half of the countries belong to the UNEP high human development group and the other half to the medium group. Informal employment is common, making workers vulnerable because of their lack of legal and social protection. The region shows high unemployment rates among youth and women. Increasing migration of female workers and highly educated people in particular to Northern America and partly to Europe is a growing concern. Common natural disasters, tropical storms, floods, earthquakes, and volcanic eruptions are also a major concern.

On the basis of high-level political commitments by ministers of labour and other high-level government representatives, and employers’ and workers’ organizations of the English- and Dutch-speaking Caribbean, the DWCPs were launched for 2010–2015 in their respective countries. In collaboration with the ILO, DWCPs have been developed in the Bahamas, Barbados, Belize, Guyana, Suriname, and the member countries of the Organization of Eastern Caribbean States (OECS), comprising the Leeward Islands: Antigua and Barbuda, St. Kitts and Nevis, Montserrat, Anguilla, and the British Virgin Islands, and the Windward Islands: Dominica, St. Lucia, St. Vincent, the Grenadines and Grenada, Martinique, and Guadeloupe. The ILO has also provided technical support to Jamaica and Trinidad and Tobago, where no formalized DWCPs have been launched.

The DWCPs were designed to enhance the ratification and reporting of ILO Conventions in order to strengthen tripartism and social dialogue, upgrade legislation, advance OSH policies, tackle discrimination on the basis of HIV/AIDS patients, and eliminate child labour. Through the programmes, measures for equalizing working conditions and formalizing the status of informal workers were developed, as well as for integrating workers with disabilities. Administrative capacities in public sector agencies and judicial capacity in the legal system were strengthened. Other inputs were the promotion of vocational training and education and the enhancement of productivity, competitiveness, employment, and trade.

According to the high-level evaluation, the sub-regional and national DW programmes were relevant, particularly in addressing youth employment, child labour, OSH, and HIV/AIDS. The evaluation team rated the overall performance of ILO’s DWCPs, strategies, and actions by triangulating the information and data gathered through desk reviews, surveys, and interviews of staff and constituents, as shown in the overall assessment graph in Figure 4 [66].

### 9.2. Evaluation of the ILO’s DWCP, Strategies and Actions in the Lower Mekong Subregion (2006–2015)

As a part of the ILO Asia Region, the subregion of the Lower Mekong countries, Cambodia, the Lao People’s Democratic Republic, Thailand, Viet Nam, and Myanmar, constitutes a relatively homogenous and closely collaborating area and one of the most rapidly developing areas in the world. The total population of the sub-region is 241 million and the workforce 130 million. Four of the Lower Mekong sub-region countries were subjected to external high-level DWCP evaluation. Myanmar has not been evaluated yet, due to the short history of DWP in the country. According to ILO reports, the countries are simultaneously experiencing several new challenges due to globalization, climate change, environmental disasters, and several traditional working life problems such as occupational injuries and diseases, hazardous physical, chemical and biological exposures, long working hours, inequalities, and psychological stress. To facilitate the positive development of working life, the ILO initiated and obtained the Lower Mekong countries’ commitment to the Decent Work Decade 2006–2015, focusing on five priority objectives: (a) sustainable productivity, (b) the youth employment challenge, (c) protecting migrant workers, (d) labour market governance, and e) local development for Decent Work. As in all DWCP evaluations, ILO-guided parameters were standardized and used in the evaluation [67].

The overall assessment on the basis of the regional evaluation is presented in Figure 5.

### 9.3. Independent Evaluation of the ILO’s DWCP, Strategies and Actions in the Western Balkans 2012–2015

In the second decade of the new millennium, working life in the Western Balkan countries of Albania, Bosnia Herzegovina, Croatia, Montenegro, North Macedonia (previously the Former Yugoslav Republic of Macedonia (FYROM)), and Serbia was exposed to several simultaneous pressures: continuing policy and structural adjustment after the socio-economic transition to market economies, the post transition economic recession followed by the 2008 global financial and economic crisis, preparation for the European Union (EU) accession, and the universal pressures of globalization. Despite the efforts of the ILO and others, the post-crisis recovery in the sub-region was considered weak due to three main problems that continued to exist after the transition and the financial crises: persistently weak economic growth, high unemployment, and weak social dialogue. In spite of these challenges, today, all the Western Balkan countries are considered upper middle-income countries according to the World Bank ranking [68].

The Western Balkan countries committed themselves to the DWCP process at different paces: Albania and Bosnia and Herzegovina developed the first generation of DWCPs between 2006 and 2007. The second generation of DWCPs was implemented between 2008 and 2011 in Albania, Bosnia and Herzegovina, Serbia, and at that time, FYROM. From 2012 onward, the third generation of DWCPs was developed in Albania and Bosnia and Herzegovina. Within the framework of the DWCPs, all five countries continued to ratify ILO conventions, which further contributed to improvements of labour or labour-related national legislation.

Taking into consideration the whole subregion, the inputs of the DWCPs were directed, in addition to the objectives of the ILO DWCP guidebook, also to recognized national needs. As a whole, the Western Balkans’ sub-regional activities covered all the four pillars of the Decent Work Agenda well, but the programme contents differed between the countries, reflecting their specific needs. Examples of activities of the different Decent Work Pillars were:(1)Rights at work: e.g., training and education in the protection of workers’ fundamental rights, introduction of models for peaceful settlements of labour disputes, promotion of gender equality at work.(2)Employment creation and enterprise development: e.g., introduction of models for employment policies and their implementation, development of employment services, local employment development, enhancement of vocational training, promotion of entrepreneurship and SMEs, and the development of labour markets, including markets for vulnerable groups.(3)Social protection: e.g., support of national OSH systems and policies and capacity building of national labour inspections, OSH training of labour inspectors and even doctors, assistance in ratification, implementation of international labour standards, advice on the development of social security, social security, and pension reforms, and capacity building.(4)Social dialogue: e.g., legal advice on and technical assistance for national social dialogue mechanisms and capacity building of social partners.

The overall performance assessment results of the evaluation are shown in Figure 6.

### 9.4. Independent High-Level Evaluation of the ILO’s Programme of Work in Four Selected Member Countries of the Southern African Development Community (SADC) (Lesotho, Madagascar, South Africa, and the United Republic of Tanzania) 2014–18

The Southern African Development Community (SADC) is well endowed with human and natural resources. Given its economic potential, the SADC is one of the most promising developing regions of the world. The economies of the SADC Member States are at different stages of development, and as a result, the state of industry varies widely throughout the region. In many Member States, agriculture plays a major role in the economy, employing almost half of the total population of the region. Mining employs just 5% of the population, but contributes 60% to the foreign exchange earnings and 10% of gross domestic product for the SADC region. Tourism is growing rapidly, employing about 11% of the workforce and contributing over 12% to GDP.

The countries of the region are diverse in their degree of development and living and working conditions. In spite of positive perspectives, the region still faces several challenges, including high levels of poverty (majority of people) and inequality, high unemployment (ranging from 4.5% to 80%) and underemployment, labour migration, and neglect of the application of law and international standards. Other challenges are low productivity and the low coverage of social security among the majority of men and women, youth, and children. For example, in South Africa, HIV/AIDS among the working-age population is high, at 12.5%.

The four countries selected for evaluation had different economic and human development indicators: Lesotho and South Africa both have middle-income status in the World Bank ranking, and HDIs of 0.518 (low) and 0.705 (medium), respectively. Madagascar and Tanzania still belong to the low-income group with low HDIs of 0.521 and 0.528, respectively [69].

The regional DWP had the following objectives and expected outcomes:

Priority 1. Regional and technical work: The priority areas had the following objectives: (a) harmonization and strengthening of functional SADC labour market information systems (LMIS), (b) development of labour migration system, and (c) elimination of human trafficking

Priority 2. Promotional work: (a) Ratification, domestication, and compliance of and compliance with International labour standards, (b) promotion of youth employment strategy, (c) compliance with SADC codes (social security, child labour, safe use of chemicals, HIV/AIDS, tuberculosis in mining), and (d) promotion of Decent Work in the informal economy in SADC Member States.

Priority 3. Information sharing: (a) improving knowledge of best practices in employment and labour policies, legislation, programmes, and social protection floors among Member States and (b) harmonization and development of skills development policies.

The inputs of the DWCPs in the four countries varied widely.

Lesotho: (a) employment creation for poverty reduction, (b) improvement of social protection coverage, and (c) development of tripartism and social dialogue.

Madagascar: The DWCP set out two priorities: (a) to promote access of vulnerable groups to employment by enhancing their employability and boosting employment generating sectors; and (b) to improve labour productivity by promoting social dialogue, fundamental principles and rights at work, as well as social protection.

Tanzania: (a) extension of social protection coverage for all; (b) promotion of the creation of productive employment; (c) improvement of compliance with labour standards and rights at work; (d) strengthening of social dialogue mechanisms at national and sectoral levels.

South Africa: (a) strengthening of fundamental principles and rights at work; (b) promotion of employment creation; (c) strengthening and broadening of social protection coverage to include vulnerable workers operating in the informal economy and informal employment; (d) strengthening of tripartism and social dialogue.

The overall assessment results of the SADC countries’ evaluation are presented in Figure 7.

### 9.5. Overall Summary of the Evaluations

The DWCP content, implementation, and actions varied between the countries, due to differences in the diagnosed country needs, national and regional circumstances, and available resources. Large numbers of region-, country-, and substance-specific conclusions and recommendations for the future development of the DWCPs were made. Some overall conclusions can also be drawn from the evaluation reports by using the ILO DWCP evaluation criteria.

The relevance of the Regional and National DWCPs was found to be high, due to the national needs identification diagnosis and participatory principle applied with all constituents’ partners. The ability of the national stakeholders to participate, however, varied widely depending on the national circumstances and resources. The DWCPs were also well synchronized with the other programmes of the UN (MDGs and SDGs) and the programmes of NGOs working for related objectives.

The programme design and coherence were ensured through effective and skilful guidance provided by the ILO, based on the ILO Decent Work Agenda, DWCP Guidebook, and guidelines and diagnostic methods prepared for regional and country levels.

Where critical conditions for full participation were met, the effectiveness and efficiency were good. In countries with lower interest or capacities for participation, capacities of institutions, national ownership of programmes, and availability of human and financial resources, the need for further development and strengthening of these resources was identified.

The impact of DWCPs in most countries was good, due to enhanced awareness of Decent Work and transfer of the DWCP objectives to national strategies.

The implementation was critically dependent on national circumstances, on political support, participation culture by stakeholders, on institutional capacities, and on human and financial resources, which in many countries need strengthening.

## 10. Discussion and Conclusions

The 2019 ILO Report, Time to Act for SDG 8, urges policy-makers around the world to help speed up progress towards SDG 8 and the implementation of the 2030 Agenda as a whole [70]. According to the Report, radical and transformative change is required across the three policy spheres of economy, society, and the environment. The report has pointed out many areas in which progress has been too slow so far, but it has also highlighted a range of opportunities for concerted and synergistic policy action. The key is to incorporate the goals of sustained growth, inclusive growth with Decent Work, and environmental integrity into a human-centred, sustainable development agenda. This is where the United Nations 2030 Agenda meets the ILO Centenary Declaration for the Future of Work. The principles and policies of Decent Work are needed also for meeting the unexpected and emerging new hazards like the COVID-19 pandemic [71,72]. Together with other UN Organizations, the ILO provides guidance and support for the world of work in the management of the global crisis. The pandemic has elevated the value of and the call for universal occupational health services into the Decent Work Agenda.

On the basis of experiences gained and the independent external evaluations performed so far, the following conclusions can be drawn:

### 10.1. Globalization

Globalization shows both positive and negative impacts on the occupational health of working people. These are not distributed equally between the countries with different degrees of development.Workers’ health and work ability and healthy and safe workplaces are factors in productive employment, sustainable economies, and overall socioeconomic development. Globalization challenges all these aspects and calls for proactive occupational health policies and actions.

### 10.2. Situation Analysis

The global analysis of health, safety, and economic burden of occupational and work-related diseases and injuries has been estimated to be at a level of 2.8 million fatalities. The majority (about 89%) of the total burden is attributed to WRDs, and about eleven percent to occupational injuries. The economic loss from such hazards amounts to 4% of GDPs on average. Recently, ILO has predicted a global loss of 25 million jobs by the COVID-19 pandemic.International surveys have shown low coverage of occupational health services (OHS) in the world (18.5% of the total global workforce). OHS are the key partner for the prevention and management of hazards for health and work ability at work and for the promotion of health and work ability. ILO C161 on Occupational Health Services aims to provide OHS for all workers.The COVID-19 pandemic has changed dramatically the perspectives for Decent Work and simultaneously demonstrated the critical value of health and safety, as well as of universal occupational health coverage in the management of the global crisis.

### 10.3. Concept and Content of Decent Work

The Decent Work Agenda of the ILO was established to equalize the impact of globalization on employment, workers’ rights, conditions of work, OSH, social protection, and social dialogue.In 2000–2019, the ILO launched a total of 121 DWCPs, i.e., in two-thirds of the 187 ILO Member States. So far, 53 DWCPs have been subjected to external evaluation, either as individual country evaluations or as a part of regional evaluations, and were deemed well guided, well documented, and well implemented.The contents of the DWCPs have been drawn up on the basis of the ILO Decent Work Agenda and the diagnosis of countries’ Decent Work needs and deficits, identified with the help of the Decent Work indicators. Thus, the programmes may differ between countries or the weight of the different Decent Work Pillars may vary between the DWCPs.

### 10.4. Occupational Health and Decent Work

The rapid changes in working life and working conditions and the parallel major demographic trends (e.g., ageing populations, rural-urban and external migration) mean that OHS must be given more emphasis. For this, the promotion, ratification, and implementation of ILO C161 on Occupational Health Services should be enhanced, aiming for universal occupational health coverage of all workers. The Basic Occupational Health Service approach (BOHS) can serve as an instrument for the implementation of UOHC.In the DWCPs, accidents and safety have been well addressed, but the health dimension and the work-related diseases, WRDs, except for HIV-AIDS at the workplace, are almost non-existent. There is a need to more firmly address the prevention and management of WRDs in DWCPs and ILO policies in general.

### 10.5. Evaluation and Information

The Decent Work concept has been deemed timely and feasible at the country level and has been widely adopted by the countries. The DWCPs are effectively guided by the ILO and designed to respond to countries’ needs (Decent Work Guidebook, Decent Work indicators, and diagnosis). They are implemented through national authorities and actors, with technical support from the ILO. The external evaluations collected data on the implementation and country experiences and provided guidance for the further development of the DWCPs.Although effective legal and technical tools, methodologies, and measures to prevent occupational accidents and diseases exist, there is a need for increased general awareness of the importance of OSH and OHS. High-level political commitment is needed for the development of national OSH systems and their effective implementation, including the development of OHS in particular.

### 10.6. Way Forward

The ILO Decent Work concept has been adopted as the key content of SDG 8 of the UN 2030 Strategy. The ILO has been assigned as a custodian of 13 SDG 8 targets in the UN Strategy. Thus, the Decent Work Agenda contributes substantially to the implementation of the UN 2030 Strategy, but needs to strengthen the implementation and particularly occupational health dimension. Through the UN SDGs and the Global Commission on the Future of Work Report, the ILO Decent Work Agenda will have a bright future at least up to 2030 and possibly beyond. This brightness is seriously shadowed by the current COVID-19 pandemic. One of the key 2030 objectives should be the UOHC of all working people of the world. This need is even more emphasized by the rapid spread of the COVID-19 pandemic, which demonstrates the unmeasurable value of occupational health and work ability of workers as a critical factor for the maintenance of the whole social fabric, health of the population, national and global economies, employment, and the overall functioning of the national, international, and global systems.

## 11. Summary

Decent Work is a unique social innovation with global coverage. The ILO has promoted and implemented it with its constituents for two decades, respecting the consistency and continuity of the ILO Decent Work Agenda. Over 120 countries have joined the ILO for implementation of the DWCPs and succeeded in integrating the DWCPs into national economic, employment, and social policies and programmes. Occupational safety and health have been included in part of, but not in all DWCPs. The external evaluations found the implementation of DWCPs on average good, but variation was wide. The external evaluation results have been used for continuous learning and improvement by all stakeholders. This has generated opportunities to achieve system-wide impact on the work life of the countries.

The sustainability of the ILO Decent Work policy was ensured through the UN 2030 Agenda, the ILO Global Commission Report on the Future of Work, and the ILO Centenary Declaration. The priority given to occupational safety and health in the DWCPs needs, however, enhancement, and the position of occupational health services needs it even more. In view of the evidence on the global burden of work-related diseases the strengthening of the occupational health approach and expansion of the coverage of occupational health services are justified according to the lines of the UN Resolution on Universal Health Coverage, including the call for Universal Occupational Health Coverage (UOHC) and the provisions of the ILO Convention No. 161. The recent pandemic emergencies further emphasize the important role of occupational health.

## Figures and Tables

**Figure 1 ijerph-17-03351-f001:**
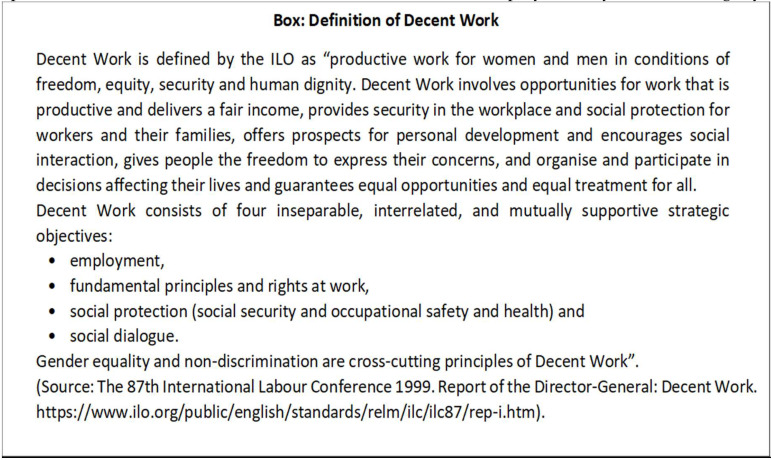
The ILO definition of decent work is collected from ILO key document to BOX [5].

**Figure 2 ijerph-17-03351-f002:**
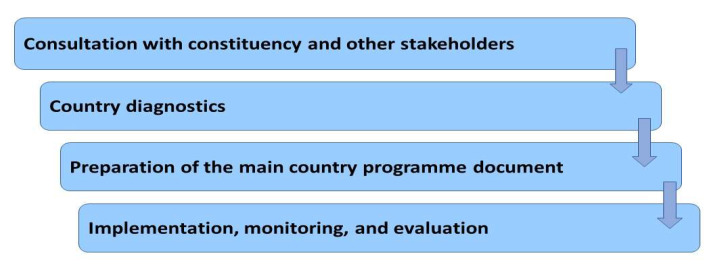
Country programming process Modified from [14].

**Figure 3 ijerph-17-03351-f003:**
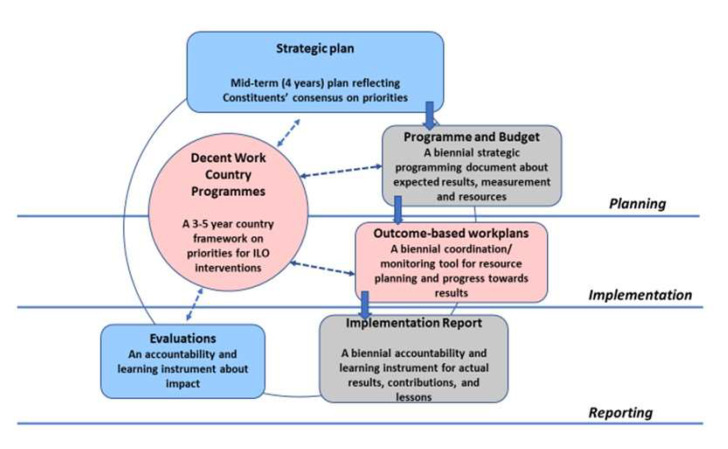
ILO results-based programming cycle. Modified from [14].

**Figure 4 ijerph-17-03351-f004:**
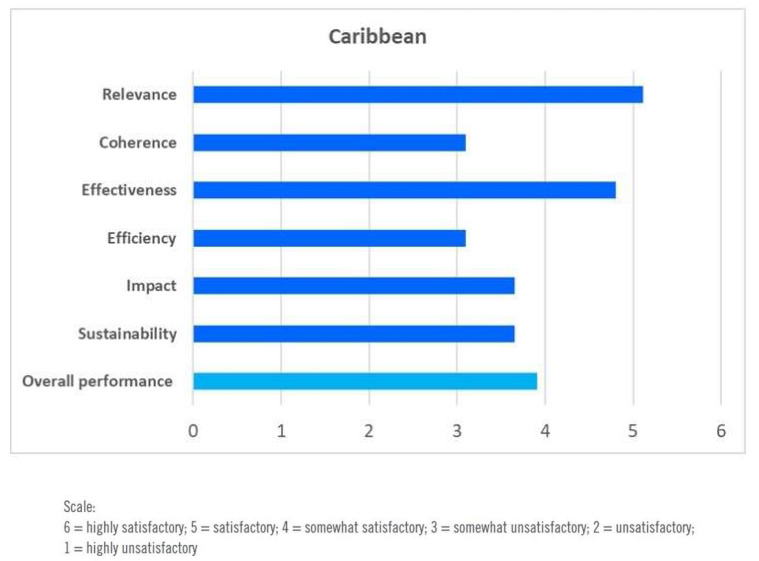
Overall assessment of DWCPs in the Caribbean. Source: [66].

**Figure 5 ijerph-17-03351-f005:**
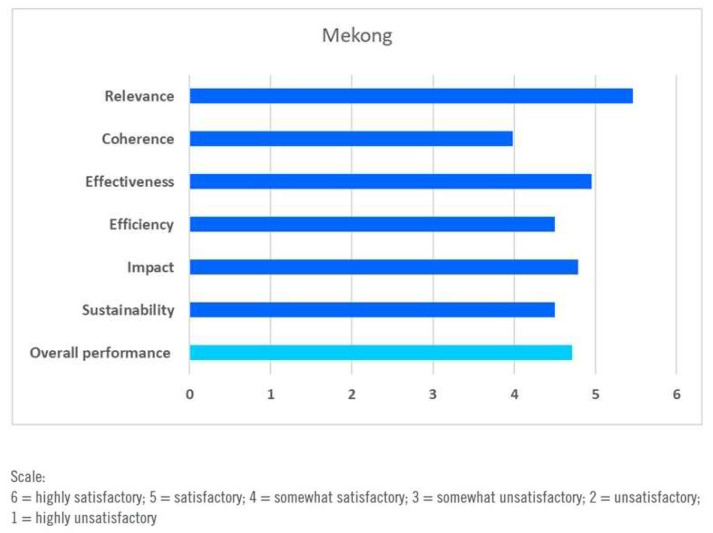
Overall assessment of DWCPs in the Lower Mekong countries. Source: [67].

**Figure 6 ijerph-17-03351-f006:**
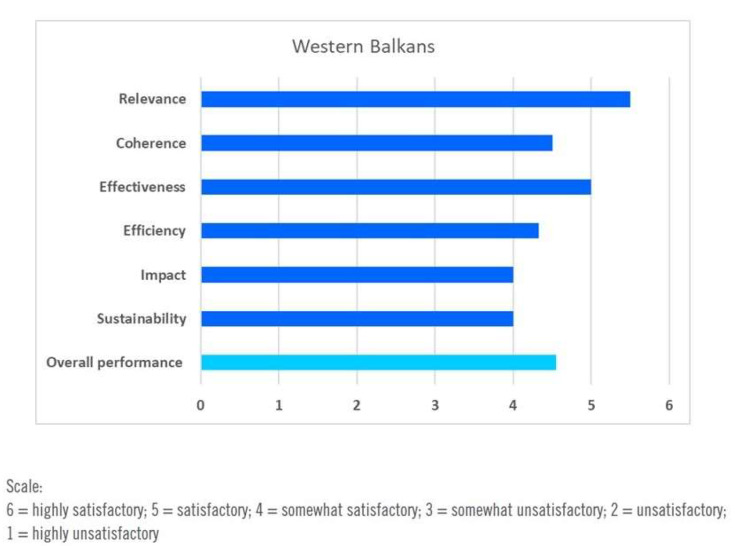
Overall performance assessment of DWCPs in the Western Balkans. Source: [68].

**Figure 7 ijerph-17-03351-f007:**
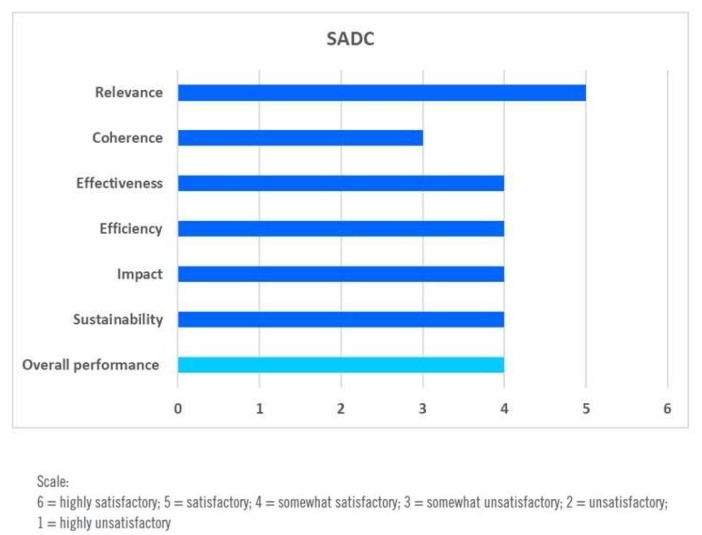
Overall performance assessment of DWCPs in the Four SADC Member Countries. Source: [69].

**Table 1 ijerph-17-03351-t001:** ILO DWCPs, 2000–2019.

ILO Region	Concept DWCP	Draft DWCP	Final DWCP	Grand Total, Countries in the Process	Total No. of Countries in the ILO Region
Africa	4	2	43	49 (91%)	54
Americas and the Caribbean	3	2	21	26 (74%)	35
Arab States	3		5	8 (67%)	11
Asia and the Pacific	2	3	19	24 (67%)	36
Europe and Central Asia			14	14 (27%)	51
Total	12	7	102	121 (64%)	187

Source: ILO Decent Work website: https://www.ilo.org/global/topics/decent-work/lang--en/index.htm. Discussions of high-level evaluations: Strategies and Decent Work Country Programmes [16].

**Table 2 ijerph-17-03351-t002:** ILO DWCPs as of 15 September 2019.

Status of Decent Work Country Programme Development by Region (as of 15 September 2019)		
Region	Draft DWCP Document (1) in the Process of Drafting and in Consultation with Tripartite Constituents	DWCP Final Document (2) Approved by Regional Director
Africa *	22	19
Arab States	1	5
Asia and the Pacific	14	15
Latin America and the Caribbean	2	3
Europe and Central Asia	2	9
Total	41	51

* In addition to DWCPs, there is one sub-regional Programme for Southern African Development Community (SADC). Development of the sub-regional DWP for the Western Africa Economic Community for West African States (ECOWAS) has commenced. Explanatory notes: (1) Draft DWCP documents include all the elements of the DWCPs developed through a process of consultation with tripartite constituents. They are appraised through the quality assurance process. (2) DWCP final documents that, while subject to modification if conditions change, have met the requirements for approval and can be cited as the vehicle for the ILO [17] (ILO 2019. Status of Decent Work Country Programme Development by Region).

**Table 3 ijerph-17-03351-t003:** Needs and provisions for OHS and DWCPs.

What Is Needed of DWCP for OHS?	What Is Needed of OHS for DWCP?
Policy containing, for example objectives for OHS National development programme for OHS Regulation on OHS Government governance, enforcement, and inspection of OHS Political will and inclusion of OHS as a priority in the national development agenda Effective feedback systems from the grassroots level to governance bodies and policies	Policy and regulatory support Institutionalization of OHS In the optimal case, provision of comprehensive, multidisciplinary content of OHS Infrastructure for universal OHS provision Competent human resources in OHS Expert support, advice, and OHS Sustainable financing for OHS Statistics and registration of occupational diseases and WRDs
**What DWCP Can Offer OHS**	**What OHS Can Offer DWCP?**
Collaboration between national health 2030 strategies and DWCPs SDG 8 Framework Tripartite support mechanism Collaboration with other activities, safety, social protection, employment Integration with the national DWCP	Occupational health dimension to DWCP and link to health sector Identification, prevention, diagnosis, and recognition of occupational and work-related diseases Provision of comprehensive OHS, including prevention, protection, care, rehabilitation, promotion of health, and work ability Provision of expert advice and services in occupational health, occupational medicine, ergonomics, psychology, rehabilitation, work ability, and compensation schemes

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
