# Peer review of "Decent Work, ILO’s Response to the Globalization of Working Life: Basic Concepts and Global Implementation with Special Reference to Occupational Health"

_ijerph, 2020, doi:10.3390/ijerph17103351_

Round 1
Reviewer 1 Report
Introduction could be improved. At the end of this paragraph you should clearly state the aim of your paper.
Methods should be improved and detailed the search strategy according to international standard of reviews. I suggest to include this citation about the relationship between decent work and occupational health: Chirico F. May the gross domestic product growth be a valid indicator of decent work? Ann Ig. 2017;29(4):332-335.Author Response
please see attachment

Reviewer 2 Report
Thank you for the opportunity to review the article.
The work deals with a very important topic, especially in the face of the current situation, which we can observe all over the world.
However, the article needs to be improved before publication.
Please find some comments and suggestions in the attached file

Round 2
Reviewer 2 Report
Thank you to the Authors for improving the article.
In scientific work everything must be precisely specified, please mention in the methodology section those important and "key documents" that were used to prepare the thesis.
Author Response
We thank again for comments and valuable recommendation for the improvement of the article.
We have complemented the article according to your recommendations as the following:
a) Added a number of references providing more information on background, ILO evaluations and methodology
b) Added Annex 2 on ILO and UN reports which were analysed for this review
c) Added Annex 3 on ILO Decent Work external evaluations in 2006-2018